# Optimization of Biomass Cultivation from *Tuber borchii* and Effect of Additives on Triterpenoid Production

Yu-Rong Fu [1,†], Parushi Nargotra [1,2,†], Chia-Hung Kuo [2,3,*] and Yung-Chuan Liu [1,*]

1   Department of Chemical Engineering, National Chung Hsing University, Taichung 402, Taiwan; fyr0707@gmail.com (Y.-R.F.); parushi11nargotra@gmail.com (P.N.)
2   Department of Seafood Science, National Kaohsiung University of Science and Technology, Kaohsiung 811, Taiwan
3   Center for Aquatic Products Inspection Service, National Kaohsiung University of Science and Technology, Kaohsiung 811, Taiwan
*   Correspondence: kuoch@nkust.edu.tw (C.-H.K.); ycliu@dragon.nchu.edu.tw (Y.-C.L.); Tel.: +886-7-3617141 (ext. 23646) (C.-H.K.); +886-4-22853769 (Y.-C.L.)
†   These authors contributed equally to this work.

**Abstract:** Edible fungi are renowned for producing biologically active secondary metabolites that possess anti-tumor activity, protect the liver and have other benefits. The cultivation of truffle mycelia through submerged fermentation has gained interest in the production of metabolites for bio-medicinal purposes. In the present study, *Tuber borchii* was cultivated by submerged fermentation to produce both biomass and triterpenoids. Various additives, including palmitic acid, stearic acid, linoleic acid, chitosan, $CaCl_2$ and limonene, were investigated to enhance triterpenoid production. It was observed that increasing the medium's linoleic acid concentration to 1 g/L increased the production of triterpenoids to 129.29 ± 6.5 mg/L, which was 2.94 times higher than the control. A number of variables, including potassium and magnesium ion concentrations and carbon and nitrogen sources and concentrations, were considered to ascertain the ideal conditions for *T. borchii* growth in submerged fermentation. The best concentrations for glucose, yeast extract, peptone, malt extract, $KH_2PO_4$ and $MgSO_4 \cdot 7H_2O$ in submerged fermentation were 19.45, 4.58, 7.91, 5.3, 0.58 and 0.82 g/L, respectively, according to response surface methodology. Validation analysis revealed that the experimental values and the predicted values were in good agreement. Under ideal circumstances, the maximum dry cell weight (2.980.18 g/L), which was 1.39 times greater than the control, was attained. Finally, the addition of 1.5 g/L linoleic acid on day 14 to the optimal medium elevated the triterpenoid production to 212.63 ± 16.58 mg/L, which was a 4.84-fold increase compared to the control.

**Keywords:** *Tuber borchii*; triterpenoids; submerged fermentation; linoleic acid; response surface methodology

## 1. Introduction

A broad category of macrofungi suitable for both culinary and therapeutic uses that are nutrient-rich are collectively referred to as edible fungi. Since ancient times, people have consumed mushrooms as a source of nourishment, and more than 2000 species of fungi, including over 200 wild species, have been discovered so far in different parts of the world, with China being the world's largest producer of edible fungi [1,2]. After plant-based and animal-based diets, edible fungi are thought to be the third major source of nutrition for humans since they almost entirely meet their nutritional needs [3]. Edible fungi have huge fruiting bodies, and apart from their culinary and organoleptic properties such as flavor, taste and texture, they are rich in carbohydrates, amino acids (alanine, asparagine, glycine and glutamine), ash content, mineral constituents, chitin and unsaturated fatty acids but lack fat content. However, the fact that they also contain a variety of bioactive substances,

such as polysaccharides (such as α- and β-glucan), proteins, peptides, polyphenols, terpenoids, vitamins and dietary fiber strengthens their medical relevance [2,4]. Moreover, it has been reported that the antibacterial, antiviral and antioxidant properties of edible fungi are mainly due to the presence of terpenoids and sterols [1].

Terpenoids are a massive and structurally diverse group of biologically active secondary metabolites that are produced by fungi to serve crucial ecological roles as signaling and defense chemicals [5]. The different known terpenoids include monoterpenes, sesquiterpenes, diterpenes, triterpenoids and steroids, which are categorized on the basis of their chemical structure. The most predominant terpenoids in medicinal fungi are triterpenoids, which are polycyclic substances generated from the cyclization of straight-chain hydrocarbon squalene and consist of 29–35 carbons [6]. Triterpenoids can be divided into linear triterpenoids and cyclic triterpenoids, based on their ring numbers. Linear triterpenoids (mainly squalene derivatives) are uncommon in medicinal fungi and act as a precursor to form cyclic triterpenoids after undergoing different structural changes, including folding, cyclization and rearrangement reactions [6,7]. The anti-cancer effect of triterpenoids is the most noteworthy of their diverse bioactive qualities, which also include liver protection, antioxidant, immunomodulatory, anti-inflammatory and anti-obesity activity. These are believed to inhibit various cancer cells by preventing nuclear factor-B (NF-B) activation, inducing natural and programmed cell death and cell cycle arrest, hindering growth, invasion, metastasis and angiogenesis to prevent and treat cancer [8,9]. Various edible and therapeutic fungi have been reported to produce triterpenoids, including *Ganoderma resinaceum* [10], *Poria cocos* [11], *Antrodia camphorate* [12], *Inonotus obliquus* [13], *Pleurotus ostreatus* [14] and *Laetiporus sulphureus* [15].

Among numerous edible fungi, truffles are well known for their major economic significance. Truffles are hypogeous fungi that belong to the genus *Tuber* and form hypogenous fruiting bodies by developing an ectomycorrhizal symbiosis with the roots of certain tree species (such as poplar, oak, willow and shrubs) [16,17]. *Tuber borchii* (previously known as *T. albidum*), a whitish truffle, is the least exploited species of truffles for the production of triterpenoids with therapeutic importance [18]. *T. borchii* is well known for its ability to produce aroma since it contains a variety of volatile chemical compounds [19,20]. Globally, all truffles are "commercially valuable delicacies" due to the presence of a peculiar taste, a high nutrient content and biologically active components. Moreover, the limited availability of naturally grown truffles values them as a luxury product [21]. The natural production of truffles has some drawbacks, such as a protracted harvesting procedure (4–12 years) and variable truffle quality due to reliance on environmental factors (temperature, soil type, and host tree) [19]. These shortcomings of natural cultivation can be overcome by employing semi-artificial cultivation using a submerged fermentation process for the growth of truffles. Submerged fermentation is a viable alternative for the effective generation of bioactive truffle mycelia and metabolites. Different *Tuber* species have been employed for the production of diverse compounds, including different volatile organic compounds, through submerged fermentation. Fermentation conditions are critically important in maintaining the formation of aroma characteristics in truffles [22,23]. However, isolation of a truffle strain is difficult from the fruiting body due to the presence of various filamentous fungi, yeast and bacteria in the fruiting body [24,25].

The production of truffle biomass through submerged fermentation is a faster process than natural cultivation. Submerged fermentation carried out under optimal environmental conditions may lead to enhanced biomass production [26,27]. It also aids in the extracellular production of biologically active secondary metabolites, enzymes and polysaccharides, which can be separated from the truffle biomass and used for further applications. Moreover, the production of mycelia biomass and bioactive compounds can be enhanced by optimizing different nutrient and environmental factors using certain statistical analyses, such as response surface methodology (RSM) [28]. Therefore, the current study reports the efficient production of both biomass and triterpenoids from *T. borchii* under submerged fermentation. The effects of different carbon and nitrogen sources on the growth of *T. borchii*

were evaluated. Different fatty acids and additives were also used as inducers to study their impact on biomass growth and triterpenoid production. Moreover, the concentration of the various medium components was also optimized using RSM in order to enhance the truffle mycelium growth. The present study is the first to report the statistical optimization of cultural variables for the production of enhanced mycelia biomass and, ultimately, triterpenoids from *Tuber borchii* under submerged fermentation.

## 2. Materials and Methods

### 2.1. Chemicals and Reagents

Chemicals, such as glucose, magnesium sulphate ($MgSO_4 \cdot 7H_2O$) and potassium dihydrogen phosphate ($KH_2PO_4$), were purchased from Showa Denko K.K, Tokyo, Japan. Bacto™ yeast extract, Bacto™ peptone, Bacto™ malt extract and vanillin were acquired from Becton, Dickinson and Company, Sparks, MD, USA and Fisher Scientific, Waltham, MA, USA, respectively. Palmitic acid, stearic acid and linoleic acid were bought from Sigma Aldrich, St. Louis, MO, USA. Chitosan, calcium chloride and 97% (R)-(+)-Limonene were obtained from Merck, Darmstadt, Germany, Avantor Performance Materials Inc., Radnor, PA, USA and Alfa Aesar, Heysham, UK, respectively. PDA, Vitamin $B_1$ and ursolic acid were procured from HiMedia, Bombay, India and Merck Ltd., Taipei, Taiwan, respectively. HCL, ethanol, glacial acetic acid and ethyl acetate were purchased from Echo Chemical Co., Ltd., Miaoli, Taiwan, and NaOH and perchloric acid were purchased from Showa kako, Co., Osaka, Japan. All chemicals used in the current study were of analytical grade.

### 2.2. Strain and Cultivation

The truffle *Tuber borchii* (ATCC® MYA1019™) used in the current study was purchased from the American Type Culture Collection (ATCC). Species identification was done using the internal transcribed spacer (ITS) gene sequence method and the sequence was submitted to GenBank under accession number AF533362. *Tuber borchii* was initially grown on potato dextrose agar medium (PDA) at 25 °C for 28 days and subcultured periodically. The cultivation was then conducted in 100 mL seed medium with a composition as shown in Table 1. Ten milliliters of seed medium was dispensed in a sterilized homogenizing bottle, and three pieces (2 × 2 cm) of truffle mycelia previously cultivated for 28 days on PDA were used to inoculate the medium. The inoculum was homogenized (V100, Osterizer, Mexico) for 10 s with a break for 5 s to obtain a uniform mycelium liquid. The homogenized mycelium culture was then transferred to the rest of the 90 mL medium and incubated at 25 °C and 100 rpm in a rotatory shaker for 14 days.

**Table 1.** Media composition.

| Components | Seed Culture Medium | Main Medium |
|---|---|---|
| | Concentration (g/L) | |
| Glucose | 5 | 20 |
| Peptone | 1 | - |
| Malt extract | - | 20 |
| Yeast extract | 1 | - |
| $MgSO_4 \cdot 7H_2O$ | 0.2 | 1 |
| $KH_2PO_4$ | 0.2 | 1 |
| Vitamin $B_1$ | - | 0.15 |

In a 500 mL Erlenmeyer flask containing 100 mL of the main culture medium, shake-flask culture was carried out. The medium composition is listed in Table 1. The pH of the medium was maintained at 7 using either 0.1 N HCl or 1 N NaOH. A 10 mL of seed broth grown for 14 days was inoculated in 90 mL main medium and incubated in a rotary shaker (100 rpm) set at 25 °C for 28 days for submerged fermentation.

### 2.3. Impact of Different Additives at Varying Concentration

The impact of six different additives, i.e., palmitic acid, stearic acid, linoleic acid, chitosan, $CaCl_2$ and limonene, on the *T. borchii* biomass and triterpenoid production was investigated. Three fatty acids, including stearic acid, palmitic acid and linoleic acid, chitosan, calcium chloride and limonene, were used in the study. In the main medium, any of the six additives was added at concentrations of 0.5, 1.0, 1.5 and 2.0 g/L. The medium was inoculated with *T. borchii* and incubated in a rotary shaker (100 rpm) set at 25 °C for 28 days for submerged fermentation. After 28 days, the culture was assessed for biomass growth and triterpenoid production.

### 2.4. Impact of Carbon and Nitrogen Sources

The effect of various concentrations of glucose on the *T. borchii* biomass and triterpenoid production was investigated. The glucose concentration in the main fermentation medium varied (5, 10, 15, 20 and 25 g/L), keeping the rest of the medium components constant, as shown in Table 1. The mycelial biomass and triterpenoid production was monitored at all glucose concentrations after 28 days of submerged fermentation. In the case of nitrogen sources, yeast extract, peptone and malt extract were used at concentrations of 5, 10, 15 and 20 g/L, and their effect on biomass growth and triterpenoid production was estimated. The carbon source concentration was kept constant.

### 2.5. Statistical Optimization for Enhanced T. borchii Biomass Production

RSM is a statistical optimization technique that includes a number of mathematical and statistical options for developing and using empirical models [27,29,30]. In the current study, the RSM-based statistical optimization for enhanced production of truffle biomass was performed in three steps, including a two-level fractional factorial design (FFD), path of steepest ascent (PSA) and central composite design (CCD). A FFD was used to identify the variables of significant importance for biomass production under submerged fermentation. It used a linear model to assess the impact of each variable on the biomass production process in further optimization. In total, six variables, i.e., glucose, yeast extract, peptone, malt extract, $KH_2PO_4$ and $MgSO_4 \cdot 7H_2O$ were chosen and expressed in coded levels where $-1$, 0 and $+1$ represented low level, middle level and high level, respectively (Table 2). The six chosen variables were employed in eight experiments, and the combined effect of variables was determined by first-order polynomial equation.

**Table 2.** Range of experimental values of response parameters in a two-level factorial design.

| Variables | Parameter | Coded Level | | |
| --- | --- | --- | --- | --- |
| | | **−1** | **0** | **+1** |
| $X_1$ | Glucose (g/L) | 5 | 15 | 25 |
| $X_2$ | Yeast extract (g/L) | 5 | 10 | 15 |
| $X_3$ | Peptone (g/L) | 5 | 10 | 15 |
| $X_4$ | Malt extract (g/L) | 5 | 10 | 15 |
| $X_5$ | $K^+$ (g/L) | 0.5 | 1 | 1.5 |
| $X_6$ | $Mg^{2+}$ (g/L) | 0.5 | 1 | 1.5 |

Furthermore, PSA was employed on the basis of the results obtained from FFD to estimate the optimal region of each variable. CCD is a rotational design that allows equal precision of estimation in all directions. CCD was employed to estimate the optimal value of each variable within the experimental range on the basis of the approximate central point of the response surface in the optimal area obtained through PSA. CCD was performed using new levels obtained from PSA. *Tuber borchii* production was fitted to a second-order polynomial equation using multiple regression analysis. An analysis of variance (ANOVA) was used to assess the fitness of the model. All experiments were designed and analyzed by the software Design Expert 6.1.

### 2.6. Assays

2.6.1. Cell Biomass

Dry cell weight (DCW) was used to express the *T. borchii* mycelial biomass. The DCW was obtained by filtering the culture broth after 28 days of submerged fermentation from Advantec No.1 filter paper with a diameter of 90 mm and a pore size of 6 µm (Toyo Roshi Kaisha Ltd., Tokyo, Japan). The cells obtained after filtration were washed thrice with deionized water and dried at 80 °C to a constant weight.

2.6.2. Total Triterpenoid Analysis

Total triterpenoid analysis was executed by dispensing 50 mg ground dried cells in 1 mL of 95% ethanol, which was extracted by incubating at 25 °C for 2 h in an ultrasonic bath (150W, D150H, Delta ultrasonic Co., Ltd., New Taipei City, Taiwan). Afterwards, 0.5 mL of the extracted sample was kept in a water bath set at 70 °C to evaporate the ethanol. The sample was then mixed with freshly prepared 5% vanillin-glacial acetic acid solution (0.2 mL) and 0.8 mL perchloric acid solution and again incubated at 70 °C in a water bath for 15 min. After incubation, the sample was cooled down in an ice bath, and 5 mL ethyl acetate was added and thoroughly mixed. The absorbance of the sample was measured by spectrophotometer (Genesys 20, Thermal Fisher Scientific Co., Ltd., Waltham, MA, USA) at a wavelength of 548 nm, and ursolic acid was used as the standard.

### 2.7. Statistical Analysis

For each test, three flasks were used each time for sampling. Each data point was expressed as a mean with a standard deviation. The Tukey test was used to compare the results with a significant difference ($p \leq 0.05$).

## 3. Results

### 3.1. Production of Tuber borchii Biomass and Triterpenoids

The growth of truffle in terms of cell dry weight and the production of total triterpenoids under submerged fermentation were estimated in the presence of glucose and malt extract as carbon and nitrogen sources. The growth curve of *T. borchii* and total triterpenoid content production are represented in Figure S1. The mycelia biomass began to increase significantly on the 13th day and reached the maximum ($2.15 \pm 0.12$ g/L) on the 28th day, after which it decreased slightly. A little growth in the mycelia biomass from 1–10 days of submerged fermentation could be due to the time taken by the in column to adapt to the new environment, indicating an adaption period [17]. The triterpenoid production pattern was associated with its growth, indicating a positive relationship between dry cell weight and triterpenoid production level. The total triterpenoid production per liter of the culture medium was maximum ($43.95 \pm 3.96$ g/L) on the 28th day after which a small decline in the triterpenoid content was observed. Therefore, in subsequent experiments, mycelia biomass and total triterpenoid content were measured after the 28th day of submerged fermentation.

### 3.2. Effects of Fatty Acids on the Production of Truffle Biomass and Triterpenoids

Fatty acids are crucial for the synthesis of various phospholipids present in cell membranes. They play a crucial role in maintaining the integrity, structure and permeability of the cell membrane [31]. Therefore, to study the impact of fatty acids on the mycelia biomass and triterpenoid content, different fatty acids (palmitic acid, stearic acid, linoleic acid) were added to the culture medium at varying concentrations. The addition of palmitic acid at various concentrations showed a little positive effect on both biomass and triterpenoid levels; however, the increased fold was less than 1.2-fold for both biomass and triterpenoid production as compared to the control (Figure 1a). During the addition of stearic acid, no significant effect was observed on either biomass or triterpenoids at any concentration (Figure 1b). For all concentrations of stearic acid, the dry cell weight and triterpenoid level remained similar to the control where no stearic acid was used. The addition of linoleic acid had an inhibitory effect on truffle growth as the dry cell weight decreased upon the addition

of linoleic acid. The cell dry weight ($0.99 \pm 0.15$ g/L) obtained at 2 g/L linoleic acid was significantly lower than that of the control ($2.15 \pm 0.12$ g/L). In contrast, the total triterpenoid content was significantly increased in the presence of linoleic acid. The presence of 1 g/L linoleic acid enhanced the triterpenoid content by 2.94 times ($129.29 \pm 6.5$ mg/L) as compared to the control ($43.95 \pm 3.96$ mg/L). However, the triterpenoid level decreased as the concentration of linoleic acid increased to 2 g/L (Figure 1c).

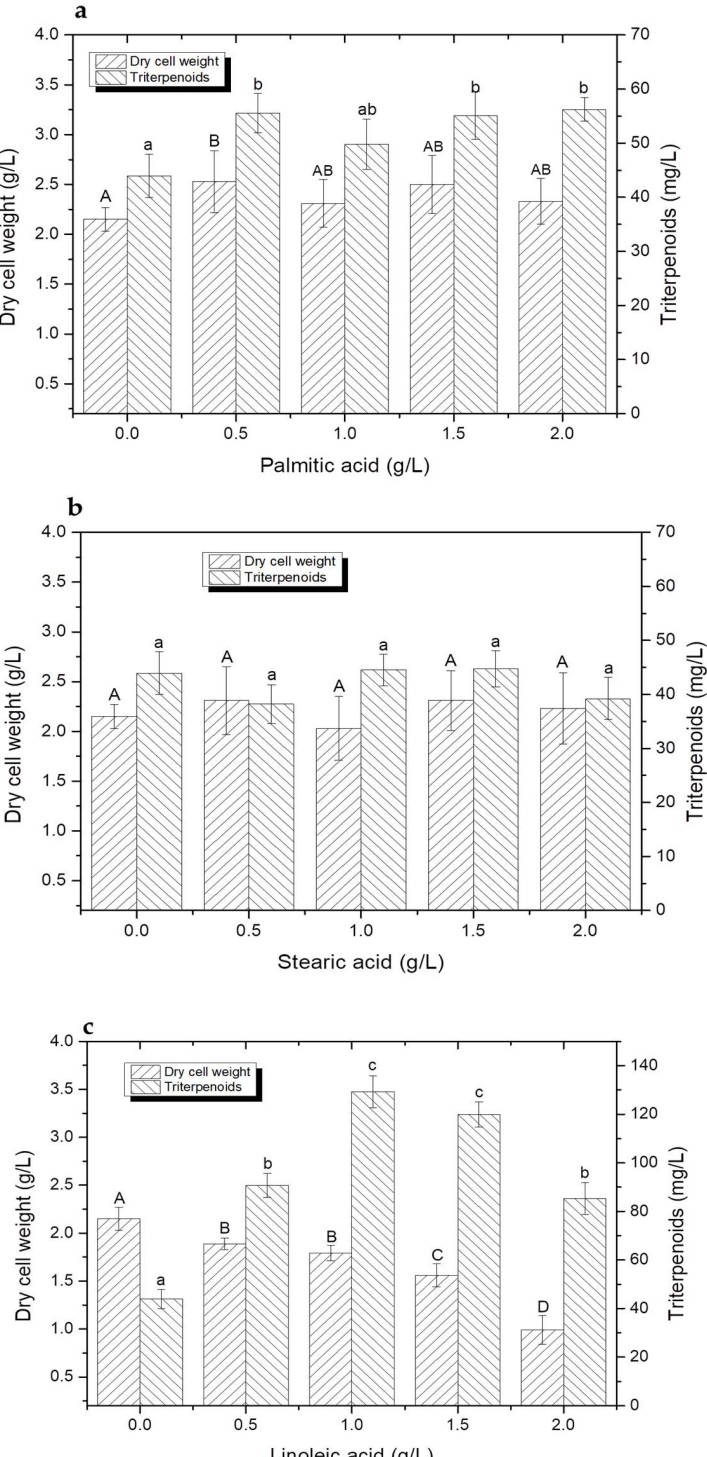

**Figure 1.** Effects of different concentrations of palmitic acid (**a**), stearic acid (**b**) and linoleic acid (**c**) on the production of *T. borchii* biomass and total triterpenoids production. Different superscripts in the uppercase indicate a significant difference ($p < 0.05$) in DCW, while different superscripts in lowercase indicate a significant difference ($p < 0.05$) in triterpenoid production.

The increase in the triterpenoid content with the addition of fatty acid could be attributed to the increased gene expression and activities of key enzymes involved in the synthesis of triterpenoids, including 3-hydroxy-3-methylglutaryl-CoA synthase (HMGS), 3-hydroxy-3-methylglutaryl-CoA reductase (HMGR), farnesyl diphosphate synthase (FPPS), squalene synthase (SQS) and squalene epoxidase (SE) [31]. The results indicated that linoleic acid increased triterpenoid production but reduced truffle growth, which could be due to the fact that biomass production is chiefly dependent on factors involved in carrying out submerged fermentation, such as the type of strain, culture composition and fermentation conditions, whereas triterpenoid production is chiefly associated with metabolic pathways [31]. Similar to our study, Huang et al. [32] reported a slight increase in the *Sanghuangporus baumii* biomass when palmitic and stearic acid were used, whereas with linoleic acid, a slight decrease in biomass was observed. However, the triterpenoid production was enhanced in the presence of linoleic acid ($510 \pm 30$ mg/L) compared to the control ($230 \pm 10$ mg/L).

### 3.3. Effects of Additives on the Production of Truffle Biomass and Triterpenoids

Elicitation plays an important role in enhancing the production of biologically active secondary metabolites. Therefore, the addition of chemical elicitors, such as chitosan, calcium chloride and limonene, as inducers at varying concentrations was investigated for their impact on the mycelial biomass and triterpenoid production. When chitosan and calcium were separately used as elicitors, the mycelial biomass remained almost constant at all concentrations, whereas the total triterpenoid content was negatively affected by their presence (Figure 2a,b). The addition of limonene was unfavorable for the growth of the truffle and total triterpenoid production. The biomass and triterpenoid levels decreased with the increase of limonene concentration (Figure 2c). In contrast to our results, the presence of chitosan, calcium chloride and limonene at an appropriate concentration has been shown to increase the triterpenoid level. It was reported by Yeh et al. [33] that chitosan and camphor extract enhanced the truffle biomass, while chitosan significantly increased triterpenoid production ($343.19 \pm 9.25$ mg/L), which was a 3.45-fold increase over that of 15 mM $CaCl_2$ ($99.41 \pm 6.99$ g/L). Liu et al. [34] reported that the best dosage of chitosan was 100 mg/L, which resulted in the increased mycelial biomass (12.3 g/L) and triterpenoid level (24.9 mg/g dry weight) from *Antrodia cinnamomea*. The concentration beyond 100 mg/L inhibited the mycelial growth, which could be due to the impact of a higher concentration of chitosan on cell wall permeability. Similarly, the calcium chloride concentration was also found to be crucial, as a higher concentration may raise the osmotic pressure of the cell wall, causing the microorganism to enter the death phase. In another study, the addition of limonene, a kind of monoterpene, at a concentration of 1% (*v/v*) resulted in maximum triterpenoid (33.93 mg/g dry weight) production from *Antrodia cinnamomea*, whereas a decrease in the triterpenoid level was observed at a concentration of 1.5 or 2% (*v/v*) [35]. Therefore, in comparing the present study results with the reported studies, it was evident that the concentration of an elicitor plays a vital role in enhancing the production of secondary metabolites.

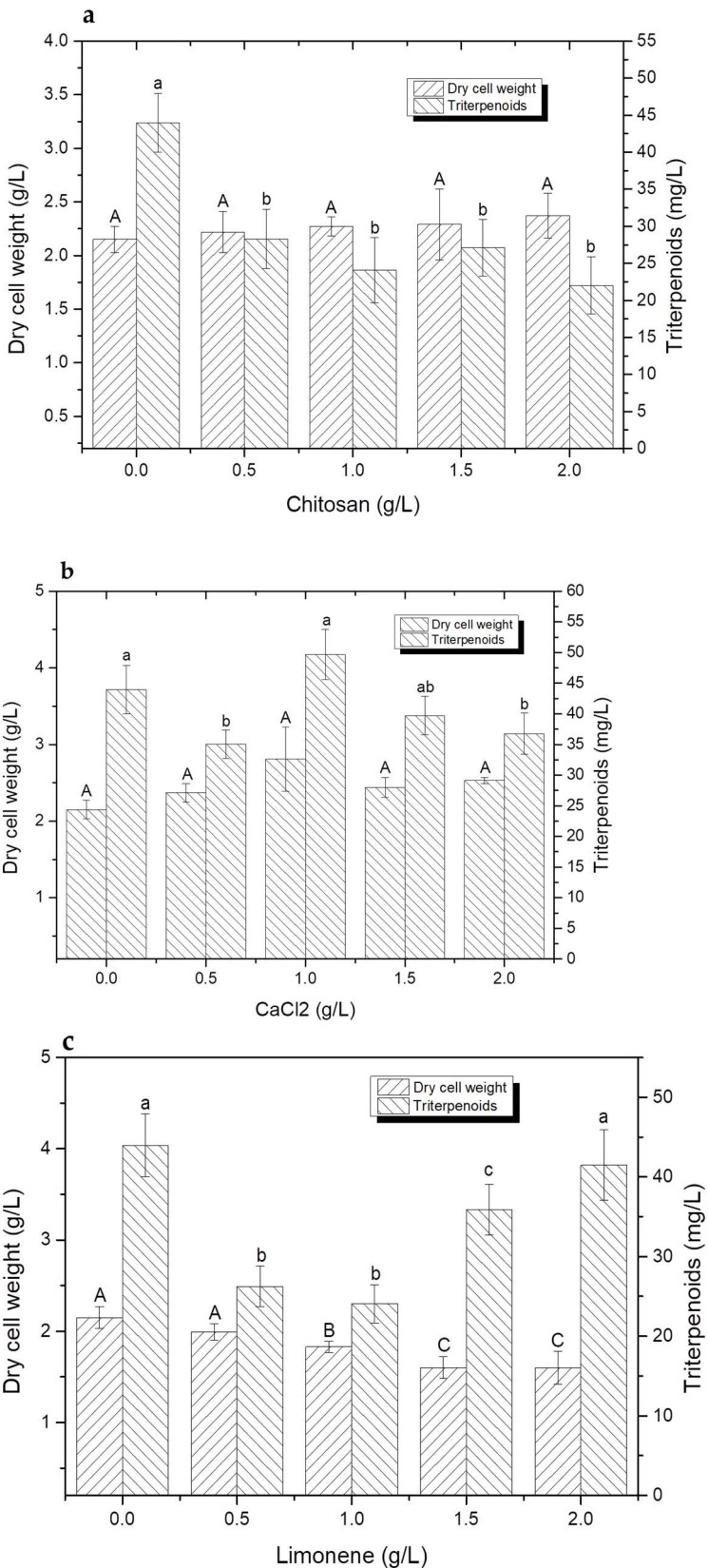

**Figure 2.** Effects of different concentrations of chitosan (**a**), calcium chloride (**b**) and limonene (**c**) on the production of *T. borchii* biomass and total triterpenoids production. Different superscripts in uppercase indicate a significant difference ($p < 0.05$) in DCW, while different superscripts in lowercase indicate a significant difference ($p < 0.05$) in triterpenoid production.

### 3.4. Effects of Carbon Sources on the Production of Truffle Biomass and Triterpenoids

Carbon sources are crucial for the growth and development of organisms. They not only provide the energy necessary for growth but also act as nutrition for metabolites or cell structures. The growth of organisms and their metabolite production greatly depends on the type of carbon source and its concentration [17]. According to the preliminary study results, where different carbon sources (glucose, fructose, sucrose and galactose) were used for the growth of *T. borchii* under solid-state fermentation, glucose was found to be the best carbon source that led to the enhanced production of mycelium. Therefore, in the current study, glucose was used at varying concentrations to study its effect on truffle biomass and triterpenoid content production under submerged fermentation. The truffle growth remained constant at all concentrations of glucose.

The triterpenoid amount produced by the truffle increased with the increasing glucose concentration after 10 g/L. As can be seen from Figure 3, a maximum triterpenoid amount of 34.94 ± 1.49 mg/L was obtained at a glucose concentration of 15 g/L, which was 1.68 times more than the triterpenoids produced at a 5 g/L concentration (20.78 ± 4.88 mg/L) and remained significantly the same at a glucose concentration of 25 g/L. Glucose is the key and first component in the synthesis of most of the compounds or metabolites in any organism. Glucose forms pyruvate as a result of glycolysis, which is then converted into acetyl CoA and subsequently enters the mevalonate pathway, where triterpenoids are biosynthesized [32,36]. The results of the present study were aligned with the previously reported study wherein glucose used at a concentration of 60 g/L yielded maximum *Sanghuangporus baumii* biomass and triterpenoid production. In another study, a high mycelial biomass of *Antrodia cinnamomea* was obtained with both maltose and glucose, whereas the maximum (31 mg/g DW) triterpenoid yield was obtained when only 2% glucose was used [37]. Therefore, the results of the present and previously reported studies thoroughly explain the importance of a carbon source in the production of biologically active metabolites.

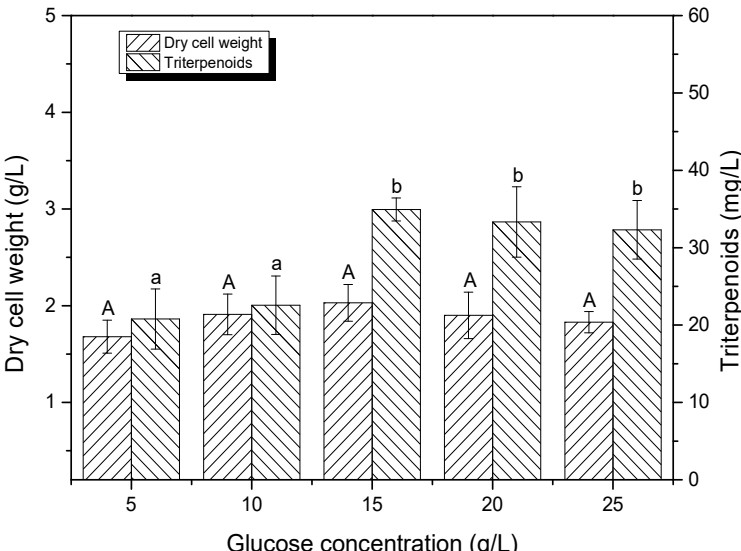

**Figure 3.** Effects of varying glucose concentrations on *T. borchii* biomass and triterpenoid production. Different superscripts in uppercase indicate a significant difference ($p < 0.05$) in DCW, while different superscripts in lowercase indicate a significant difference ($p < 0.05$) in triterpenoid production.

### 3.5. Effects of Nitrogen Sources on the Production of Truffle Biomass and Triterpenoids

Nitrogen sources are essential for fungi to synthesize amino acids, proteins, nucleic acids and cytoplasm [17]. Nitrogen sources can not only promote the growth of mycelium but also influence the shape of mycelium. In the preliminary analysis, yeast extract was found to be the best nitrogen source for the growth of mycelium, as it resulted in high

mycelium density and diameter under solid state fermentation for 28 days. This was followed by peptone and malt extract, although the mycelial density was sparse, but the growth rate was fast. Therefore, these three nitrogen sources were selected and used at different concentrations for the growth of *T. borchii* biomass and triterpenoid production under submerged fermentation. The results indicated that with the increase in yeast extract concentration from 5 g/L to 15 g/L, mycelium biomass was also increased. The highest cell dry weight of $2.9 \pm 0.18$ g/L was achieved when 15 g/L yeast extract concentration was used, which was 1.46 times more than the biomass at 5 g/L ($1.98 \pm 0.04$ g/L) (Figure 4a). The maximum total triterpenoids amount of $47.57 \pm 2.27$ mg/L was also obtained when yeast extract was used at 15 g/L concentration, which was 1.56 times higher than the amount found at 5 g/L yeast extract ($30.41 \pm 4.85$ mg/L) (Figure 4a).

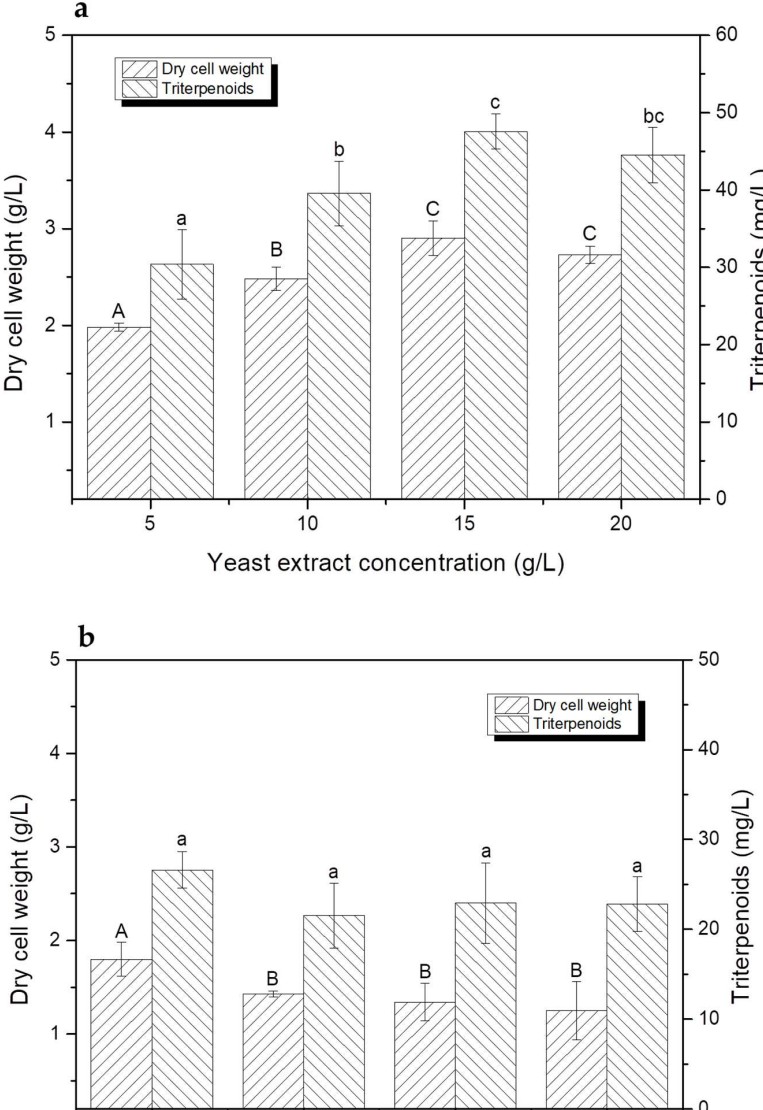

**Figure 4.** *Cont.*

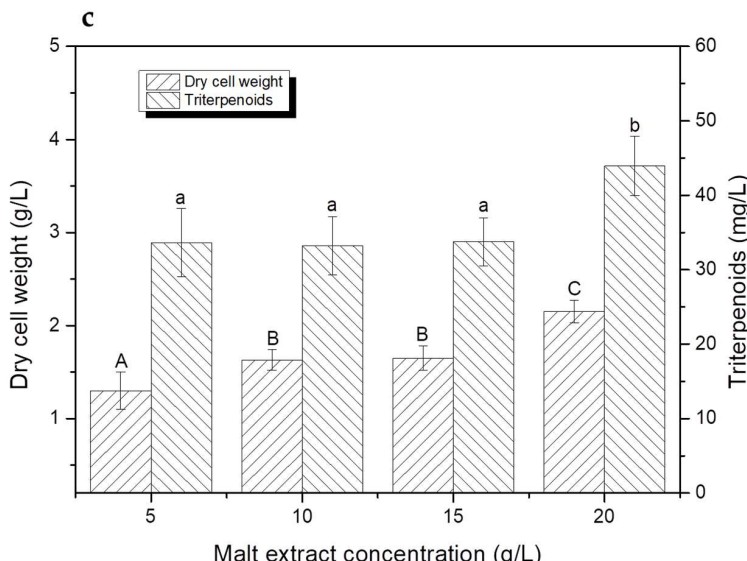

**Figure 4.** Effect of varying concentrations of yeast extract (**a**), peptone (**b**) and malt extract (**c**) on *T. borchii* biomass and triterpenoid production. Different superscripts in uppercase indicate a significant difference ($p < 0.05$) in DCW, while different superscripts in lowercase indicate a significant difference ($p < 0.05$) in triterpenoid production.

In the case of peptone, the increase in concentration beyond 5 g/L had a negative impact on the growth of the truffle. In comparison to mycelial biomass of 1.25 g/L which was obtained at 20 g/L peptone concentration, the mycelial biomass obtained at a 5 g/L peptone concentration was 1.44 times (1.80 g/L) greater (Figure 4b). The presence of certain growth inhibitors in peptone may be the cause of the inhibition of truffle development at higher peptone concentrations, where peptone's inhibitive effects outnumber its stimulating benefits [38]. The triterpenoid production was not affected by the increase in the concentration of peptone and remained stable at all concentrations (Figure 4b). When malt extract was used as a sole nitrogen source, the highest biomass ($2.1 \pm 0.23$ g/L) and triterpenoids amount ($43.95 \pm 3.96$ mg/L) was obtained at a concentration of 20 g/L. The minimal mycelium growth ($1.3 \pm 0.24$ g/L) was observed at 5 g/L, which was 1.62 times lower than the biomass obtained at 20 g/L (Figure 4c). Various nitrogen sources have been examined for biomass and triterpenoid production from different edible fungi. Nitrogen sources (malt extract, yeast extract and corn steep powder) were investigated for their impact on cell growth and triterpenoid production by *A. cinnamomea* CCRC36716. A maximum cell growth and triterpenoid production of $12.52 \pm 0.03$ g/L cell dry weight and 30 mg/g DW were obtained, respectively, when 3% corn steep powder was used as a nitrogen source [37]. In another study, the submerged culture of *Hypocreales* sp. NCHU01 was used to establish a method for producing truffle endophytic fungus triterpenoids. The results showed that triterpenoid production was increased to high levels of 327.49 mg/L and 343.16 mg/L by the presence of sucrose and yeast extract, respectively [33].

### 3.6. Statistical Optimization for the Production of Truffle Biomass

Any industrial process development strategy always prioritizes process variable optimization [29,39,40]. The RSM-based statistical tool was used to optimize the conditions for the enhanced growth of truffles and subsequently high triterpenoid production. Earlier in the current study, the media components were optimized using one variable at a time. However, unlike the one variable at a time method, RSM takes into consideration the combined effects of all components when other elements are arbitrarily kept constant [41]. Therefore, for better accuracy and to deeply comprehend the effect of media components on truffle biomass and triterpenoid production, a two-level fractional factorial design, steepest ascent method and RSM (Design Expert 6.1) was used. Six variables inducing glucose, yeast extract, peptone,

malt extract, monopotassium phosphate ($KH_2PO_4$) and magnesium sulphate ($MgSO_4$) were used in the two-level fractional factorial design. A two-level fractional factorial design was used to find the positive direction (or slope) for each variable. Their coded terms are given in Table 3 and the regression equation obtained is given below as Equation (1).

$$DCW = 1.84 + 0.058\,X_1 - 0.18\,X_2 - 0.014\,X_3 - 0.078\,X_4 + 0.007042\,X_5 - 0.054\,X_6 \quad (1)$$

**Table 3.** Results of a two-level fractional factorial design.

| Experiment No. | $X_1$ | $X_2$ | $X_3$ | $X_4$ | $X_5$ | $X_6$ | Dry Cell Weight (g/L) |
|---|---|---|---|---|---|---|---|
| 1 | −1 | −1 | −1 | +1 | +1 | +1 | 2.12 |
| 2 | +1 | −1 | +1 | −1 | +1 | −1 | 2.84 |
| 3 | −1 | −1 | +1 | +1 | −1 | −1 | 2.34 |
| 4 | +1 | +1 | +1 | +1 | +1 | +1 | 1.72 |
| 5 | −1 | +1 | +1 | −1 | −1 | +1 | 1.75 |
| 6 | −1 | +1 | −1 | −1 | +1 | −1 | 2.11 |
| 7 | +1 | −1 | −1 | −1 | −1 | +1 | 2.47 |
| 8 | +1 | +1 | −1 | +1 | −1 | −1 | 2.03 |

In order to determine the best direction for changing the levels of the investigated variables, the path of the steepest ascent (PSA) approach was used. The process of heading gradually along the path toward the optimal concentration region is known as the steepest ascent path. When the effect value of the variable is positive as per the regression equation obtained from two-level fractional factorial, a high value is chosen, while when the effect value is negative for a variable, a low value is selected. Here, as per Equation (1), only variables $X_1$ and $X_5$ had a positive impact on the growth of the truffle biomass. PSA was applied to use the optimal locus to reach the best location of each variable. Based on Equation (1), the slope of the regression equation was followed by the direction along the steepest path, and the new units added to the original base points were estimated following the method in Table 4. In Table 4, the center point of the two-level fractional factorial experiment is used as the base point of the PSA method. The difference between the extreme values of each variable and the center point is represented by unit, whereas slop signifies the steepness of each variable, i.e., it is the coefficient of each variable in Equation (1). Furthermore, the proportion is the product of unit and slope and multiplying proportion by q factor (q = 2) results in a new unit, i.e., the new steep ascent path. Therefore, as per the calculation from Table 4, a total of five sets of experiments were generated by adding/subtracting the new unit to/from the base points. An increase in the mycelial biomass was observed until trial 3, which resulted in the highest mycelial growth (2.4 ± 0.02 g/L), and afterwards, the mycelial growth decreased. Moreover, it was also inferred that the optimal value for the enhanced biomass was near the domain of experiment 3. Hence, the central point for the central composite design was reallocated to the following conditions: glucose 18 g/L, yeast extract 4.5 g/L, peptone 9 g/L, malt extract 8 g/L, $KH_2PO_4$ 1.0 g/L, $MgSO_4$ 0.8 g/L.

**Table 4.** Experimental design and results of steepest ascent path method.

| | $X_1$ | $X_2$ | $X_3$ | $X_4$ | $X_5$ | $X_6$ | DCW (g/L) |
|---|---|---|---|---|---|---|---|
| (1) Base point | 15 | 10 | 10 | 10 | 1 | 1 | |
| (2) Unit | 10 | 5 | 5 | 5 | 0.5 | 0.5 | |
| (3) Slop | 0.058 | −0.18 | −0.014 | −0.078 | 0.007042 | −0.054 | |
| (4) Proportion = (2) × (3) | 0.58 | −0.9 | −0.07 | −0.39 | 0.003521 | −0.027 | |
| (5) New unit = (4) × 2 | 1.16 | −1.8 | −0.14 | −0.78 | 0.007042 | −0.054 | |
| Expt. No. 1 | 16.16 | 8.2 | 9.86 | 9.22 | 1.007042 | 0.946 | 2.05 |
| Expt. No. 2 | 17.32 | 6.4 | 9.72 | 8.44 | 1.014084 | 0.892 | 2.18 |
| Expt. No. 3 | 18.48 | 4.6 | 9.58 | 7.66 | 1.021126 | 0.838 | 2.4 |
| Expt. No. 4 | 19.06 | 2.8 | 9.44 | 6.88 | 1.024647 | 0.811 | 2.21 |
| Expt. No. 5 | 20.22 | 1 | 9.3 | 6.1 | 1.031689 | 0.757 | 2.21 |

The six variables with new levels were set as shown in Table 5 to establish the optimal conditions for the enhanced production of truffle biomass. The results of 33 experiments in total are shown in Table 6 and the regression equation is expressed as Equation (2). The variables for which the probability value (*p*-value) was less than 0.05 were considered significant. Thus, only the significant variables, i.e., $X_2$, $X_5$, $X_1^2$, $X_2^2$, $X_3^2$, $X_4^2$, $X_5^2$, $X_6^2$, $X_1X_4$, $X_1X_6$, $X_2X_5$, $X_2X_6$, $X_3X_4$, $X_3X_5$, $X_4X_6$ and $X_4X_6$, were shown in Equation (2).

$$\begin{aligned} \text{Dry cell weight (g/L)} = 2.94 + 0.048X_2 - 0.10X_5 - 0.032X_1^2 - 0.20X_2^2 - 0.13X_3^2 - \\ 0.053X_4^2 - 0.090X_5^2 - 0.19X_6^2 - 0.062X_1X_4 + 0.046X_1X_6 + 0.044X_2X_5 + 0.063X_2X_6 + \\ 0.083X_3X_4 + 0.032X_3X_5 + 0.096X_4X_5 + 0.039X_4X_6 \end{aligned} \quad (2)$$

**Table 5.** Range of experimental values of independent variables of central composite design.

| Variables | Parameter | Coded Level | | |
|---|---|---|---|---|
| | | −1 | 0 | +1 |
| $X_1$ | Glucose (g/L) | 16 | 18 | 20 |
| $X_2$ | Yeast extract (g/L) | 2 | 4.5 | 7 |
| $X_3$ | Peptone (g/L) | 6 | 9 | 12 |
| $X_4$ | Malt extract (g/L) | 4 | 8 | 12 |
| $X_5$ | $K^+$ (g/L) | 0.5 | 1 | 1.5 |
| $X_6$ | $Mg^{2+}$ (g/L) | 0.4 | 0.8 | 1.2 |

**Table 6.** Results of the central composite design for enhanced *T. borchii* biomass production.

| Expt. No. | $X_1$ | $X_2$ | $X_3$ | $X_4$ | $X_5$ | $X_6$ | Dry Cell Weight (g/L) |
|---|---|---|---|---|---|---|---|
| 1 | −1 | −1 | −1 | +1 | +1 | −1 | 2.14 |
| 2 | 0 | 0 | 0 | 0 | 0 | 0 | 2.94 |
| 3 | −1 | −1 | −1 | −1 | −1 | −1 | 2.68 |
| 4 | −1 | +1 | +1 | +1 | +1 | −1 | 2.51 |
| 5 | 0 | 0 | 0 | 0 | 0 | 0 | 2.87 |
| 6 | +1 | +1 | −1 | +1 | −1 | +1 | 2.31 |
| 7 | 0 | 0 | 0 | 0 | +2 | 0 | 2.38 |
| 8 | +1 | −1 | +1 | −1 | +1 | +1 | 1.89 |
| 9 | 0 | 0 | −2 | 0 | 0 | 0 | 2.48 |
| 10 | 0 | 0 | 0 | 0 | −2 | 0 | 2.79 |
| 11 | 0 | 0 | 0 | 0 | 0 | 0 | 3.02 |
| 12 | −1 | −1 | +1 | +1 | +1 | +1 | 2.08 |
| 13 | +1 | −1 | −1 | +1 | −1 | −1 | 2.22 |
| 14 | +1 | +1 | +1 | −1 | +1 | −1 | 2.09 |
| 15 | −1 | −1 | +1 | −1 | −1 | +1 | 2.04 |
| 16 | 0 | 0 | 0 | 0 | 0 | −2 | 2.2 |
| 17 | 0 | −2 | 0 | 0 | 0 | 0 | 2.05 |
| 18 | 0 | 0 | +2 | 0 | 0 | 0 | 2.35 |
| 19 | +1 | +1 | +1 | +1 | −1 | −1 | 2.1 |
| 20 | −1 | +1 | −1 | +1 | +1 | +1 | 2.3 |
| 21 | +2 | 0 | 0 | 0 | 0 | 0 | 2.87 |
| 22 | +1 | −1 | −1 | −1 | +1 | −1 | 2.33 |
| 23 | 0 | 0 | 0 | 0 | 0 | 0 | 2.99 |
| 24 | 0 | +2 | 0 | 0 | 0 | 0 | 2.24 |
| 25 | +1 | −1 | +1 | +1 | −1 | +1 | 2.18 |
| 26 | 0 | 0 | 0 | 0 | 0 | 0 | 2.91 |
| 27 | −2 | 0 | 0 | 0 | 0 | 0 | 2.76 |
| 28 | 0 | 0 | 0 | +2 | 0 | 0 | 2.68 |
| 29 | −1 | +1 | +1 | −1 | −1 | −1 | 2.32 |
| 30 | −1 | +1 | −1 | −1 | −1 | +1 | 2.45 |
| 31 | 0 | 0 | 0 | −2 | 0 | 0 | 2.78 |
| 32 | +1 | +1 | −1 | −1 | +1 | +1 | 2.31 |
| 33 | 0 | 0 | 0 | 0 | 0 | +2 | 2.15 |

Analysis of variance (ANOVA) on the RSM model was carried out to assess the fitness for the polynomial model Equation (2). The ANOVA results indicated that the RSM model was significant with a *p* value of 0.0003 and the coefficient of determination $R^2$ = 0.9957 authenticated the good correlation between experimental and predicted values. The polynomial model was further differentiated, and using a statistical program, a maximum value was estimated to determine the ideal condition. The optimal conditions are shown in Figure 5. The ideal concentration predicted by the RSM software was glucose 19.45 g/L, yeast extract 4.58 g/L, peptone 7.91 g/L, malt extract 5.3 g/L, $KH_2PO_4$ 0.58 g/L and $MgSO_4 \cdot 7H_2O$ 0.82 g/L. This showed a predicted optimal biomass of 3.075 g/L, which was 1.43 times more than the control group. The experiment was conducted in triplicate under the aforementioned medium composition to validate the results obtained by the RSM model. A maximum dry cell weight of 2.98 ± 0.18 g/L was achieved on day 28 of the submerged fermentation under optimal concentration of media components, which was in close proximity to the predicted biomass and was 1.39 times higher than the control (2.15 ± 0.12 g/L) (Figure S2). This outcome suggested that the model was properly applied to explain the correlation between the media component concentration and mycelial growth. Optimization of media components for the better growth and production of metabolites from various edible fungi has been attempted previously; however, this is the first study to report the optimization of cultural variables for the enhanced growth of *T. borchii*. Si et al. [41] attempted the RSM-based three-step optimization (Plackett–Burman, PSA and Box–Behnken design) for enhanced exopolysaccharide production from *Ganoderma lingzhi*. The optimization resulted in a 3.16-fold higher exopolysaccharide yield compared to the basal medium alone. In another study, the cultivation conditions for triterpenoid production from *Antrodia cinnamomea* under submerged fermentation were optimized using RSM. Glucose and yeast extract were found to be the best carbon and nitrogen sources and the yield of triterpenoids was enhanced by 75.23% under optimized conditions compared to before optimization [42]. The results indicate that the optimization of cultivation variables is essential to achieve high biomass/metabolite yield.

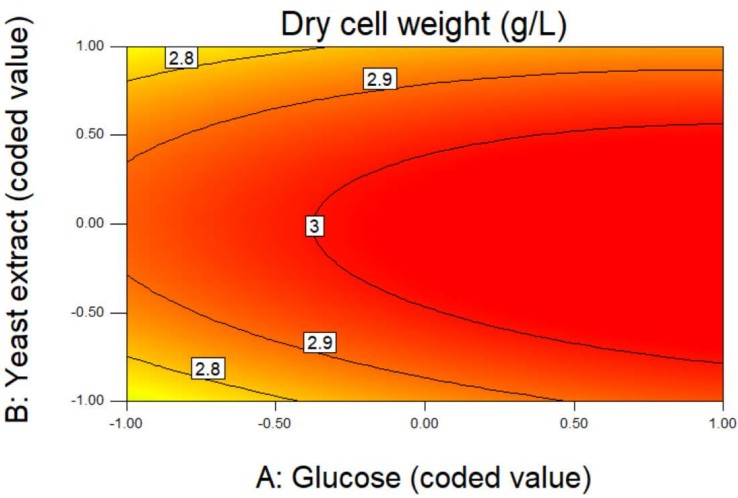

**Figure 5.** The count plot shows the effect of glucose and yeast extract on the optimal *T. borchii* biomass production at peptone 7.91 g/L, malt extract 5.3 g/L, $KH_2PO_4$ 0.58 g/L and $MgSO_4 \cdot 7H_2O$ 0.82 g/L. Numbers on contours denote dry cell weight (g/L).

### 3.7. Effects of Additives on Truffle Biomass and Triterpenoids Produced in Optimized Media

According to the results of the previously mentioned additive experiments, the addition of linoleic acid led to an enhanced total triterpenoid concentration. Therefore, the production of biomass and triterpenoids was attempted with the addition of linoleic acid in the optimal media composition. The results indicated no significant difference in the mycelial biomass and triterpenoid level between the control and optimized media in any concentration of linoleic acid. Moreover, in the case of mycelial biomass, a decrease in growth was observed with the increasing concentration of linoleic acid (Figure 6a). Since the addition of linoleic acid hampered the growth of the truffle even in the optimized media from day one, the impact of adding linoleic acid on the 14th day of submerged fermentation was investigated. The rationale behind adding linoleic acid on the 14th day was that the truffle is already in the exponential phase of growth and must have produced substantial mycelial biomass. It was found that the addition of 1.5 g/L triterpenoid on the 14th day resulted in a maximum

total triterpenoid amount of 212.63 ± 16.58 g/L, which was 4.84 times more than the control group (43.95 ± 3.96 g/L). The triterpenoid level remained almost constant, even at 2 g/L concentration. Moreover, the biomass yield also remained stable in all concentrations of linoleic acid and was higher than the biomass yield obtained upon adding linoleic acid on the 0th day (Figure 6b). From these results, it may be inferred that the optimal strategy for producing triterpenoids in a submerged culture of *T. borchii* may be to consume glucose at first for biomass growth and then to utilize linoleic acid for triterpenoid production.

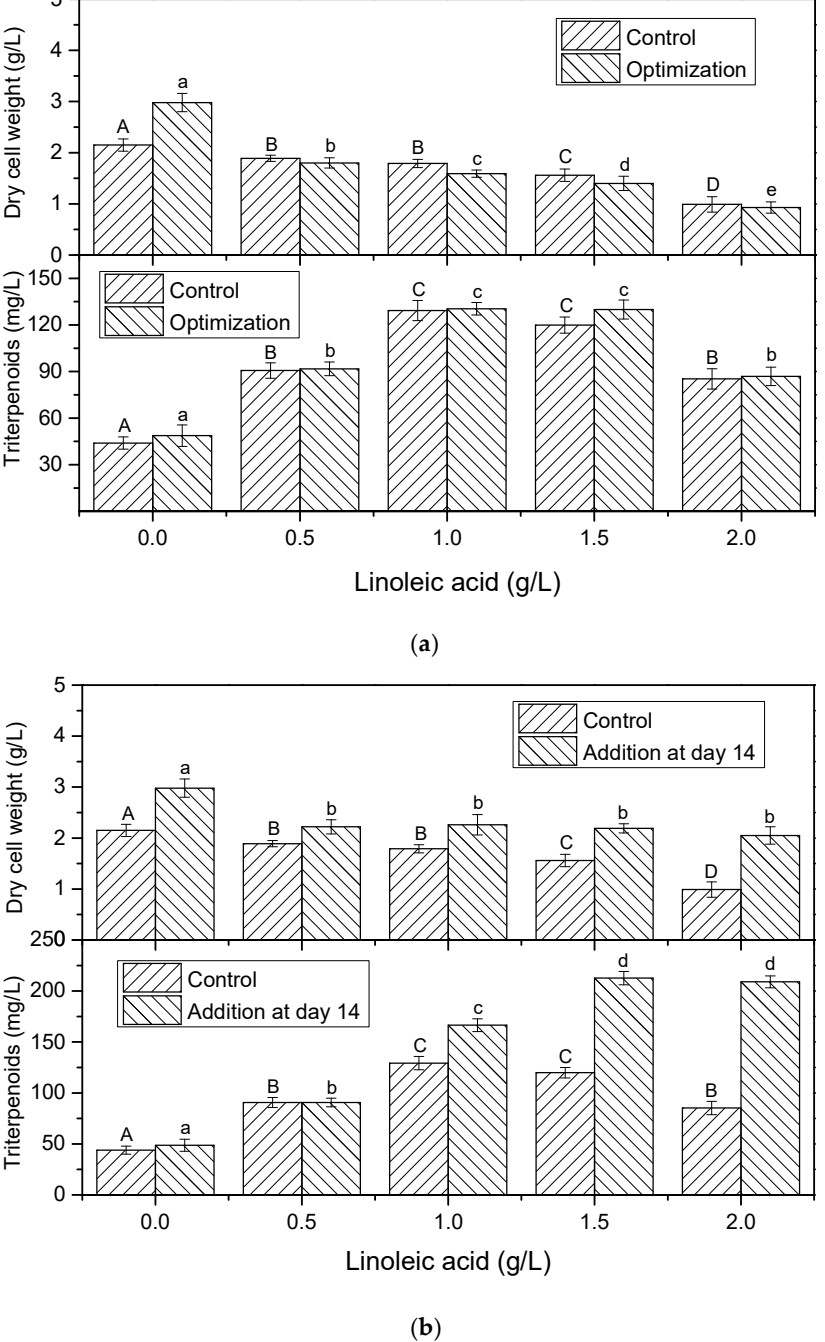

(**a**)

(**b**)

**Figure 6.** Effect of different concentrations of linoleic acid on *T. borchii* biomass and triterpenoid production grown in optimized media; added on 0th day (**a**), added on 14th day (**b**). Different superscripts in uppercase indicate a significant difference ($p < 0.05$) between control media (unoptimized media) at various concentrations of linoleic acid for both DCW and triterpenoid production, while different superscripts in lowercase indicate a significant difference ($p < 0.05$) between optimized media at various concentrations of linoleic acid for both DCW and triterpenoid production.

## 4. Conclusions

The current study is the first ever report on the RSM-based optimization of media composition for enhanced production of *T. borchii* biomass, followed by triterpenoid production. The focus of this work was to improve the triterpenoid production from *T. borchii* grown in optimized media composition in the presence of additives. The supplementation of various additives indicated that linoleic acid was the most effective in increasing the total triterpenoid level ($129.29 \pm 6.5$ mg/L), which was 2.94 times higher than the control group ($43.95 \pm 3.96$ mg/L) when truffles were grown in unoptimized media composition. The three-step statistical optimization significantly boosted mycelial biomass production ($2.98 \pm 0.18$ g/L) and was in close proximity to the predicted biomass amount by the software. Moreover, the addition of linoleic acid on the 14th day of submerged fermentation using optimized media also resulted in a maximum total triterpenoids content of $212.63 \pm 16.58$ mg/L. The presence of chitosan, calcium chloride and limonene had a negative impact on triterpenoid production. However, the reason behind this remains unclear from the current study. Therefore, further analysis of changes in the gene expression of key enzymes involved in triterpenoid synthesis may aid in understanding the impact of selective additives on triterpenoid production from this strain. The most important limitation of this study is that, when the process goes into scale-up, the oxygen mass transfer and the shear stress are quite different from those at the flask level. Therefore, the biomass growth would not be the same as in the flask. In order to perform the scale-up of this process, the parameters in the bioreactor should be obtained when it comes to process realization in industrial applications. The triterpenoids produced by the truffles can be further assessed for various biological activities, such as anti-cancer, antioxidant, immunomodulatory, anti-inflammatory and anti-obesity activity. Additionally, *T. borchii* may be further investigated for the synthesis of various sterols and phenolic compounds, followed by analysis of their bioactivities.

**Supplementary Materials:** The following supporting information can be downloaded at: https://www.mdpi.com/article/10.3390/fermentation9080735/s1, Figure S1: Growth curve of *T. borchii* and total triterpenoid production under submerged fermentation; Figure S2: Growth curve of T. *borchii* grown in optimized media under submerged fermentation.

**Author Contributions:** Conceptualization, Y.-C.L.; methodology, Y.-R.F.; validation, P.N.; formal analysis, Y.-R.F.; investigation, Y.-R.F.; writing—original draft preparation, P.N., C.-H.K. and Y.-C.L.; writing—review and editing, P.N., C.-H.K. and Y.-C.L.; supervision, Y.-C.L. All authors have read and agreed to the published version of the manuscript.

**Funding:** This work was supported by research-funding grants provided by the National Science Council of Taiwan, R.O.C. (Grant no. NSTC-112-2811-E-005-009) and in part by the Ministry of Education, Taiwan, R.O.C. under the ATU plan.

**Institutional Review Board Statement:** Not applicable.

**Informed Consent Statement:** Not applicable.

**Data Availability Statement:** Data are contained within the article.

**Acknowledgments:** Author PN thankfully acknowledges the National Science and Technology Council of Taiwan, R.O.C., for providing a Post-Doctoral fellowship.

**Conflicts of Interest:** The authors declare no conflict of interest.

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
