# Peer review of "Optimization of Biomass Cultivation from Tuber borchii and Effect of Additives on Triterpenoid Production"

_fermentation, doi:10.3390/fermentation9080735_

Round 1
Reviewer 1 Report
The manuscript titled ‘Optimization of biomass cultivation from Tuber borchii and effect of additives on triterpenoids production’ has been reviewed. The article is well-written and original, but some changes are necessary before publishing in Foods journal.
There are some articles regarding the fermentation process in truffle samples that are not referenced. I recommend including a paragraph about the fermentation process in truffles in the introduction section. Examples:
· Liu, R. S., Jin, G. H., Xiao, D. R., Li, H. M., Bai, F. W., & Tang, Y. J. (2015). Screening of the key volatile organic compounds of Tuber melanosporum fermentation by aroma sensory evaluation combination with principle component analysis. Scientific reports, 5(1), 17954.
· Tang, Y. J., Liu, R. S., & Li, H. M. (2015). Current progress on truffle submerged fermentation: a promising alternative to its fruiting bodies. Applied microbiology and biotechnology, 99, 2041-2053.
· Tang, Y., Li, Y. Y., Li, H. M., Wan, D. J., & Tang, Y. J. (2011). Comparison of lipid content and fatty acid composition between Tuber fermentation mycelia and natural fruiting bodies. Journal of agricultural and food chemistry, 59(9), 4736-4742.
· Li, Y. Y., Wang, G., Li, H. M., Zhong, J. J., & Tang, Y. J. (2012). Volatile organic compounds from a Tuber melanosporum fermentation system. Food chemistry, 135(4), 2628-2637.
· Liu, R. S., Zhou, H., Li, H. M., Yuan, Z. P., Chen, T., & Tang, Y. J. (2013). Metabolism of L-methionine linked to the biosynthesis of volatile organic sulfur-containing compounds during the submerged fermentation of Tuber melanosporum. Applied microbiology and biotechnology, 97, 9981-9992.
T. borchii was previously called T. albidum. Please check the manuscript published for further information about these truffles.
Table 1 and 2 could be combine and reduce the number of figure/tables.
I recommend carrying out a multivariable study (PCA) with the different data of triterpenoids when the additives are included.
Could you include the 2D RSM graphic?
Why didn´t you check the phenolic compounds and sterols since they have anti-inflammatory, antiobesity, anticancer, and antioxidant bioactivities?
None
Author Response
Reviewer 1
Comment: The manuscript titled ‘Optimization of biomass cultivation from Tuber borchii and effect of additives on triterpenoids production’ has been reviewed. The article is well-written and original, but some changes are necessary before publishing in Fermentation journal.
There are some articles regarding the fermentation process in truffle samples that are not referenced. I recommend including a paragraph about the fermentation process in truffles in the introduction section. Examples:
- Liu, R. S., Jin, G. H., Xiao, D. R., Li, H. M., Bai, F. W., & Tang, Y. J. (2015). Screening of the key volatile organic compounds of Tuber melanosporum fermentation by aroma sensory evaluation combination with principle component analysis. Scientific reports, 5(1), 17954.
- Tang, Y. J., Liu, R. S., & Li, H. M. (2015). Current progress on truffle submerged fermentation: a promising alternative to its fruiting bodies. Applied microbiology and biotechnology, 99, 2041-2053.
- Tang, Y., Li, Y. Y., Li, H. M., Wan, D. J., & Tang, Y. J. (2011). Comparison of lipid content and fatty acid composition between Tuber fermentation mycelia and natural fruiting bodies. Journal of agricultural and food chemistry, 59(9), 4736-4742.
- Li, Y. Y., Wang, G., Li, H. M., Zhong, J. J., & Tang, Y. J. (2012). Volatile organic compounds from a Tuber melanosporum fermentation system. Food chemistry, 135(4), 2628-2637.
- Liu, R. S., Zhou, H., Li, H. M., Yuan, Z. P., Chen, T., & Tang, Y. J. (2013). Metabolism of L-methionine linked to the biosynthesis of volatile organic sulfur-containing compounds during the submerged fermentation of Tuber melanosporum. Applied microbiology and biotechnology, 97, 9981-9992.
Author response: Thank you so much for your valuable comments and suggestions. As per your comment, we have now cited references 1 and 2 (suggested by you) in the revised manuscript. However, we have not cited other references, since we felt only 1-2 references listed above were enough to cite the literature on submerged fermentation for Tuber growth. Moreover, due to the comment to reduce Introduction from another reviewer, we have added only relevant information about submerged fermentation of Truffle in the revised MS.
Comment: T. borchii was previously called T. albidum. Please check the manuscript published for further information about these truffles.
Author response: It is right that T. borchii was previously called T. albidum. We have now added its previous name in the introduction section of the revised MS, along with a citation. However, due to the comment to reduce Introduction from another reviewer, we have not added any additional information and retained only relevant information.
Comment: Table 1 and 2 could be combine and reduce the number of figure/tables.
Author response: We have combined Table 1 and 2 and added few figures in a supplementary file.
Comment: I recommend carrying out a multivariable study (PCA) with the different data of triterpenoids when the additives are included.
Author response: We did not have sufficient data for PCA analysis due to the lack of triterpenoid compounds composition via LC-mass analysis. However, we aim to carry out this study in future.
Comment: Could you include the 2D RSM graphic?
Author response: We have added a new figure of 2D RSM graph as suggested.
Comment: Why didn´t you check the phenolic compounds and sterols since they have anti-inflammatory, antiobesity, anticancer, and antioxidant bioactivities?
Author response: We aim to test the presence of bioactive compounds such as polyphenols and sterols for their anti-inflammatory, antiobesity, anticancer, and antioxidant bioactivities in future work experiments. The same has been added as the future perspective in the conclusion section.

Reviewer 2 Report
Overall, this research manuscript is readable and meaningful. Some comments and suggestions are provided for authors' reference to revise the manuscript.
1. Specify the additives in the manuscript;
2. American type Culture Collection (ACTT) should be American type Culture Collection (ATCC);
3. More discussion with supporting references should be supplemented;
4. Limitations and future prospects should be added at the end of mainbody or conclusion part for helping future research.
5. A graphical abstract can be provided for quickly understanding the main idea of this manuscript.
May check the entire manuscript again during revision stage.
Author Response
Review 2
Comment: Overall, this research manuscript is readable and meaningful. Some comments and suggestions are provided for authors' reference to revise the manuscript.
Author response: Thank you for your response and appreciation for the manuscript. We have addressed all your comments and made changes accordingly in the revised Manuscript (MS).
Comment 1. Specify the additives in the manuscript;
Author response: The different additives used in the study, such as palmitic acid, stearic acid, linoleic acid, chitosan, CaCl2, and limonene, have now been mentioned and specified in the 2.3 section of the methods.
Comment 2. American type Culture Collection (ACTT) should be American type Culture Collection (ATCC);
Author response: The abbreviation of American type Culture Collection has been corrected to ATCC in the revised MS.
Comment 3. More discussion with supporting references should be supplemented;
Author response: As per your suggestion, we have now discussed our results with the previously reported studies in the literature. However, since there are fewer reports available on the optimization of medium conditions for enhanced T. borchii biomass production and subsequently triterpenoid production, we have discussed the results of the present study with the most related previous studies.
Comment 4. Limitations and future prospects should be added at the end of mainbody or conclusion part for helping future research.
Author response: The limitations and future prospects of the present study have now been added in the conclusions section of the revised MS to make it more informative and is as follow
The presence of chitosan, calcium chloride and limonene had a negative impact on triterpenoid production. However, the reason behind this remains unclear from the current study. Therefore, further analysis of changes in the gene expression of key enzymes involved in triterpenoids synthesis may aid in understanding the impact of selective additives on triterpenoid production from this strain. The most important limitation of this study is that when the process goes into scale-up, the oxygen mass transfer and the shear stress are quite different from those at the flask level. Therefore, the biomass growth would not be the same as in the flask. In order to perform scale-up of this process, the parameters in the bioreactor should be obtained when it comes to process realization in industrial applications. The triterpenoid produced by the truffles can be further assessed for various biological activities such as anti-cancer, antioxidant, immunomodulatory, anti-inflammatory, and anti-obesity. Additionally, T. borchii may further be investigated for the synthesis of various sterols and phenolic compounds, followed by analysis of their bioactivities.
Comment 5. A graphical abstract can be provided for quickly understanding the main idea of this manuscript.
Author response: We have added a graphical abstract as suggested.
Comments on the Quality of English Language
Comment: May check the entire manuscript again during revision stage.
Author response: As per your suggestion, we have now revised the whole manuscript and checked for English language and grammatical errors.

Reviewer 3 Report
An interesting manuscript with exciting results, if everything could be perfect and the data will be 100% applied in practice. The most important limit of the manuscript is the lack of explain/describe its limits (if any). Is it possible to have not any limit and the experiments (that are not exhaustive in number) conduct to perfect and definitive conclusions?
The introduction should be re-organized and eventually reduced. Some data are known from other articles, some other data are not the most prominent from other articles (i.e. reference 4 with another subject and another study goal that what was cited in the present manuscript) (i.e. reference 22 that is referring itself to other cited articles 32, 33, 34 discussing about the contamination of a truffle strain). Please re-think the conclusions and do not end with „etc”.
Typos and English topic should be checked again (e.g. please use 25degreeC written in the same mode, every time).
Even PSA could represent an abbreviation for „path of steepest ascent”, it is confusing with PSA abbreviation most used in medicine all over the world (maybe it could be better to not abbreviate).
Author Response
Review 3
Comments and Suggestions for Authors
Comment: An interesting manuscript with exciting results, if everything could be perfect and the data will be 100% applied in practice. The most important limit of the manuscript is the lack of explain/describe its limits (if any). Is it possible to have not any limit and the experiments (that are not exhaustive in number) conduct to perfect and definitive conclusions?
Author response: Thank you for appreciating the manuscript (MS) and providing valuable comments to improve its quality. We have now added the limitations and future prospects of the present study in the conclusion section of the revised MS and read as follow
The presence of chitosan, calcium chloride and limonene had a negative impact on triterpenoid production. However, the reason behind this remains unclear from the current study. Therefore, further analysis of changes in the gene expression of key enzymes involved in triterpenoids synthesis may aid in understanding the impact of selective additives on triterpenoid production from this strain. The most important limitation of this study is that when the process goes into scale-up, the oxygen mass transfer and the shear stress are quite different from those at the flask level. Therefore, the biomass growth would not be the same as in the flask. In order to perform scale-up of this process, the parameters in the bioreactor should be obtained when it comes to process realization in industrial applications. The triterpenoid produced by the truffles can be further assessed for various biological activities such as anti-cancer, antioxidant, immunomodulatory, anti-inflammatory, and anti-obesity. Additionally, T. borchii may further be investigated for the synthesis of various sterols and phenolic compounds, followed by analysis of their bioactivities.
Comment: The introduction should be re-organized and eventually reduced. Some data are known from other articles, some other data are not the most prominent from other articles (i.e. reference 4 with another subject and another study goal that what was cited in the present manuscript) (i.e. reference 22 that is referring itself to other cited articles 32, 33, 34 discussing about the contamination of a truffle strain). Please re-think the conclusions and do not end with „etc”.
Author response: Thank you for your valuable comment to improve the introduction section of the MS. We have revised and reduced the Introduction section wherever possible. We also removed the least related references as suggested and cited the content related to the contamination of a truffle strain with a more relevant reference. Moreover, we have also deleted the word ‘etc.’ from the end of the conclusion and revised the conclusion in the MS.
Comments on the Quality of English Language
Comment: Typos and English topic should be checked again (e.g. please use 25degreeC written in the same mode, every time).
Author response: We have checked and uniformly added 25 °C in the revised manuscript.
Comment: Even PSA could represent an abbreviation for „path of steepest ascent”, it is confusing with PSA abbreviation most used in medicine all over the world (maybe it could be better to not abbreviate).
Author response: The abbreviation ‘PSA’ used in the present manuscript is clearly related to the term ‘path of steepest ascent’ and, therefore, may not create any confusion in the reader’s mind. We have retained the abbreviation since this word is often used in MS.
